# Metabolic Modifications in Terpenoid and Steroid Pathways Triggered by Methyl Jasmonate in *Taxus* × *media* Hairy Roots

**DOI:** 10.3390/plants11091120

**Published:** 2022-04-20

**Authors:** Katarzyna Sykłowska-Baranek, Monika Kamińska, Cezary Pączkowski, Agnieszka Pietrosiuk, Anna Szakiel

**Affiliations:** 1Department of Pharmaceutical Biology and Medicinal Plant Biotechnology, Faculty of Pharmacy, Medical University of Warsaw, 1 Banacha Street, 02-097 Warsaw, Poland; katarzyna.syklowska-baranek@wum.edu.pl (K.S.-B.); agnieszka.pietrosiuk@wum.edu.pl (A.P.); 2Department of Plant Biochemistry, Faculty of Biology, University of Warsaw, 1 Miecznikowa Street, 02-096 Warsaw, Poland; mkaminska@biol.uw.edu.pl (M.K.); myhacp@biol.uw.edu.pl (C.P.)

**Keywords:** elicitation, hairy roots, methyl jasmonate, steroids, *Taxus* × *media*, triterpenoids

## Abstract

The in vitro cultures of *Taxus* spp. were one of the first plant in vitro systems proved to exert the positive effect of elicitation with methyl jasmonate (MeJA) on the biosynthesis of specialized metabolites. The main aim of the present study is to examine the effect of MeJA treatment on the steroid and triterpenoid content of two genetically different hairy root lines of *Taxus* × *media*, KT and ATMA. The results revealed that the two lines differed in the total content of steroids and triterpenoids (in the ATMA root line, their amounts were lower than those in the KT line by 43% and 30%, respectively), but not in the composition of these compounds. The metabolic response to elicitation with MeJA was different: in the KT root line, the content of steroids decreased by 18%, whereas it increased by 38% in the ATMA line. Several metabolic features were common, including the characteristic changes in the ratio of sitosterol to stigmasterol content, caused by the very sharp boost in stigmasterol levels, the increase in the amount of glycoside forms of sterols, as well as in triterpenoid and total phenolic content. It is the first report on modifications of the terpenoid biosynthetic pathway in *Taxus* hairy root cultures triggered by MeJA, concerning steroids and triterpenoids.

## 1. Introduction

Plant in vitro cultures, including hairy roots (hairy root cultures, HRCs), have been widely applied in biotechnology to obtain highly valuable plant-derived products with various biological and pharmacological activities [1,2]. To achieve the enhanced productivity of the desired phytochemicals, several strategies have been developed for the stimulation of specific biosynthetic pathways [3,4]. Among these strategies, elicitation is one of the most useful and favored biotechnological tools, appreciated for combining a practical ease with a high effectiveness [5,6]. Regarding biotic elicitors, phytohormone jasmonic acid (JA) and its derivative, methyl jasmonate (MeJA), have become the most commonly applied due to their demonstrated enhancing effects on the production of numerous plant specialized metabolites, e.g., alkaloids, terpenoids and phenylpropanoids [5,7,8].

One of the first in vitro systems proved to exert the positive result of the MeJA treatment on the biosynthesis of pharmacologically active compounds was the in vitro cultures of *Taxus* spp. producing paclitaxel and related taxanes [5,9,10]. Paclitaxel (trade name Taxol), a tricyclic diterpene alkaloid with the molecular formula C_47_H_51_NO_14_, originally isolated from the bark of the pacific yew tree (*Taxus brevifolia* Nutt.), was reported to be a very promising anticancer drug in 1964, and it is still one of the most effective and widely applied anticancer plant-derived drugs [11,12,13]. Although the elicitation of *Taxus* in vitro cultures with MeJA has been considered particularly effective for boosting taxane production, it was often simultaneously observed that this positive effect was accompanied with the hampering of biomass formation [9].

Paclitaxel is derived from the universal diterpene precursor, geranylgeranyl diphosphate (GGPP), and phenylalanine. Its basic carbon skeleton is synthesized by the isoprenoid pathway, so it can be assumed that paclitaxel biosynthesis may compete with other pathway branches, leading to various subclasses of terpenoids (Figure 1).

Decreasing the carbon flux through competitive pathways has been shown to be one of the useful approaches to improve the production of desired specialized metabolites [14]. Therefore, since the biosynthesis of paclitaxel and squalene-derived products (steroids and triterpenoids) theoretically share the isoprenoid pathway upon farnesyl diphosphate (FPP), already during the first experiments on the stimulation of paclitaxel production it was assumed that the downregulation of squalene synthase, or any other inhibition of steroid and triterpenoid biosynthesis, may be an advantageous strategy [15,16,17,18]. However, the stimulation of paclitaxel biosynthesis, obtained as a result of application of squalene synthase inhibitors (e.g., squalestatine [17]) or growth retardants (such as chlorocholine chloride (CCC) [15,16,18]), was not as significant as demonstrated in other methods, such as precursor-feeding or elicitation. Usually, the enhancement of the paclitaxel content in *Taxus* spp. in vitro cultures does not account for more than a two-fold in the case of CCC [16,18] and three-fold in the case of squalestatine [17]. Moreover, particularly after elicitation with MeJA, the content of steroids in plant in vitro cultures of various plant species is often reported to be diminished, thus the effect of the reduced carbon flow in the biosynthetic pathway of these compounds seems to be achieved without any external inhibitors [4,7]. In turn, in some studies on *Taxus* in vitro cultures exposed to shear stress, specific modifications of steroid contents have been reported, indicating that these compounds play an important role in the response to stress factors [19].

The main aim of the present study is to examine the effect of the MeJA treatment on the steroid and triterpenoid content of two genetically different hairy root lines of *Taxus* × *media*: the KT line, obtained by transformation with the *Agrobacterium rhizogenes* strain LBA 9402 [20], and the ATMA line, which carries taxadiene synthase (*T**XS)* transgene from *T. baccata* [21]. Taxadiene synthase catalyzes the first committed step in the synthesis of paclitaxel and its congeners, i.e., the cyclization of GGPP. Although this enzyme does not seem to be rate-limiting in taxane production, its presence (accompanied by *ROL* genes of *A. rhizogenes*) has led to significantly higher levels of taxanes in comparison with lines transformed only by the *A. rhizogenes* wild strain [10]. As it was previously reported, the ATMA and KT lines differed in their potential for taxane accumulation, secretion and taxane profile [13]. To our knowledge, the present study is the first report on the modifications of the steroid and triterpenoid metabolism in *Taxus* hairy root cultures triggered by MeJA. The obtained results, combined with the analysis of the total content of phenolic compounds, provides new data on the influence of the MeJA treatment on the interplay among various metabolic pathways, and they might contribute to a better understanding of the relations between general and specialized metabolism in elicited plant in vitro cultures.

## 2. Results

### 2.1. The Identification of Steroids and Triterpenoids in T. × media Hairy Roots

The two lines of *T.* × *media* hairy roots (KT and ATMA) applied in the present study were shown to accumulate nine steroid compounds in a free form: five sterols and four steroid ketones. The identified sterols were: campesterol (24*R*-ergost-5-en-3β-ol), cholesterol (cholest-5-en-3β-ol), isofucosterol (24*Z-*stigmasta-5,24(28)-dien-3β-ol, synonym: Δ5-avenasterol), sitosterol (stigmast-5-en-3β-ol) and stigmasterol (22*E*-stigmasta- 5,22-dien-3β-ol). Sterols were accompanied by four steroid ketones: campestenone (5-campestenone, synonym: 24-methylcholest-5-en-3-one), sitostenone (4-stigmasten- 3-one), stigmastane-3,6-dione and tremulone (stigmasta-3,5-dien-7-one). All analyzed sterols and steroids were identified according to their basic MS spectra (obtained without derivatization); the identification was additionally supported by the comparison of their retention time and chromatographic mobility to the respective parameters of available authentic standards, as well as the comparison with data from MS libraries and the literature (see Section 4.7, Appendix A). No biosynthetic intermediates (e.g., cycloartenol or 24-methylenecycloartanol) or saturated sterols (e.g., campestanol or sitostanol) were found in the analyzed extracts, at least in detectable amounts. The fractions of sterol esters and glycosides, analyzed after the alkaline or acidic hydrolyses of the appropriate fractions (Section 4.5 and Section 4.6), contained all the sterols previously identified in a free form, i.e., campesterol, cholesterol, isofucosterol, sitosterol and stigmasterol.

In addition to sterols and steroid ketones, in both *T.* × *media* hairy root lines, the two most commonly occurring triterpenoid alcohols (monols) were found, i.e., α-amyrin (urs-12-en-3β-ol) and β-amyrin (olean-12-en-3β-ol), as well as the corresponding acids, i.e., ursolic acid (3β-hydroxy-urs-12-en-28-oic acid), found also in a form of naturally occurring methyl ester, and oleanolic acid (3β-hydroxy-olean-12-en-28-oic acid).

The structures of all identified steroids, organized in a biosynthetic order, are presented in Figure 2. The basic structures of the identified triterpenoids are presented in Figure 3, and their retention times and MS spectra are included in Appendix A.

### 2.2. The Effect of the MeJA Treatment on Steroid and Terpenoid Metabolism in the KT Line of T. × media Hairy Roots

Elicitation with MeJA did not qualitatively change the composition of the sterols and steroid ketones identified in a free form in respective fractions obtained chromatographically from the diethyl ether extracts of KT line hairy roots (Section 4.4); however, it significantly influenced the proportions between individual compounds, particularly sitosterol and stigmasterol. This phenomenon could be noticed clearly on the obtained chromatograms (Figure 4). Sitosterol was the predominant compound in control (untreated) roots, whereas its peak at retention time (Rt) of 36.4 min (peak number 4) decreased after elicitation, while the peak of stigmasterol (Rt 34.5 min, number 3) increased dramatically.

Apart from the visible difference in the ratio of sitosterol to stigmasterol, numerous peaks appeared on the chromatograms of the elicited KT root line (Figure 4, KT MeJA chromatogram) in a retention time range from 7 to 29 min, which were either not present or very small on the chromatograms of the fraction from the control KT root line (KT control). The majority of these peaks were associated with C15 and C20 alcohols or ketones, with some of them being identified tentatively (based on data from Wiley-NIST) as sesquiterpenoid or diterpenoid alcohols, e.g., albicanol (Rt 9.2 min), corymbolone (Rt 11.7 min) or ferruginol (Rt 18.8 min). In turn, the peak of Rt 18 min was tentatively identified as retinol, and the peak of Rt 20.2 min as retinol acetate, pointing to the presence of derivatives of carotenoids (indeed, the occurrence of yellow pigments could be noticed in the extracts obtained from the MeJA-elicited KT root line).

The quantification of steroids is presented in Table 1. The results include not only free forms of the identified compounds, but also sterols present as esters and glycosides. As it could be assumed from the peak height on the chromatograms, sitosterol was the most predominant compound among steroids occurring in the free form in the control KT root line, constituting 80% of sterols and 68% of the total steroid fraction. The second abundant was campesterol (11% of sterols). Steroid ketones constituted 14% of the total steroid fraction, with a predominating amount of sitostenone. The amount of sterols occurring in ester and glycosidic forms did not exceed 1.8% of the total content of sterols. In the control KT roots, sitosterol was the main sterol occurring in both ester and glycoside forms; however, the proportions among the other sterols, released from their conjugated forms, did not reflect exactly the proportions among the free sterols, e.g., stigmasterol was the second abundant among esters and the less abundant among glycosides.

The total content of steroids decreased by 18% in the KT root line treated with MeJA; however, as it was mentioned previously, it was not the most striking phenomenon concerning the sterol profile in the elicited roots. The most dramatic change was the exponential increase in stigmasterol content, more than 10-fold. Thus, the ratio of sitosterol to stigmasterol equaled 21:1 in the control KT line, whereas it was 1:1.15 after elicitation. Meanwhile, the content of other sterols decreased, i.e., sitosterol by 2.4-fold, cholesterol by 2.5-fold, isofucosterol by 5-fold and campesterol only by 18%. The content of steroid ketones also decreased by 18%; the only exception within this fraction was sitostenone, increasing by 10%. The influence of the MeJA treatment on the content of sterols occurring in ester and glycoside forms was different. The content of esters decreased by 42%, whereas the content of glycosides increased 2.5-fold. Among esters, stigmasterol was the only compound whose level did not decrease after elicitation with MeJA, i.e., it remained practically the same as in untreated roots. In turn, the content of stigmasterol in the form of glycosides boosted eight-fold, whereas the increase in the content of the other sterols did not exceed two-fold.

In comparison to the content of steroids, the content of triterpenoids in the KT root line was low (Table 2).

In both the control and elicited hairy roots, the most abundant compound was the methyl ester of ursolic acid (51% of the total triterpenoid fraction in control roots, 48% in the MeJA-treated samples), followed by its precursor, α-amyrin. Thus, ursane-type triterpenoids were more abundant than oleanane-type compounds. Free acids (oleanolic and ursolic) were the minor compounds, together constituting only 8% of the total triterpenoid fraction in the control roots and 11% in elicited roots. Elicitation with MeJA increased the total content of triterpenoids almost two-fold, practically uniformly for all the analyzed compounds.

### 2.3. The Effect of MeJA Treatment on Steroid and Terpenoid Metabolism in the ATMA Line of T. × media Hairy Roots

As in the KT line, no compositional changes were noticed in the steroid fractions of the ATMA root line after elicitation with MeJA; furthermore, a similar influence on the proportion between sitosterol and stigmasterol was visible on the obtained chromatograms (Figure 5), although the peak of stigmasterol did not predominate the peak of sitosterol.

As previously, additional peaks appeared on the chromatogram of the elicited ATMA root line (Figure 5, ATMA MeJA chromatogram); however, they seemed to be less numerous and smaller than in the case of the KT MeJA chromatogram. The majority of these peaks were associated with C16, C18 and C20 alcohols, ketones and acids, tentatively identified as, e.g., hexadecanol (Rt 10.9 min), octadecanol (Rt 14.4 min) and, as identified previously in the KT root line, retinol (Rt 18 min) and retinol acetate (Rt 20.2 min).

The quantification of steroids occurring in the ATMA root line is presented in Table 3. The total content of steroids (including esters and glycosides) in the control ATMA hairy roots was markedly lower (by 43%) than in the control KT roots. Again, sitosterol was the prevailing compound (82% of sterols, 62% of the total steroid fraction), followed by campesterol (12% of sterols). Steroid ketones constituted 21% of the total steroid fraction, with a predominating amount of sitostenone, as in the KT root line. The amount of sterols occurring in ester and glycoside forms constituted 2.8% of the total content of sterols. The proportion between these two sterol forms was different in the control ATMA root line than the control KT line, i.e., the amount of glycosides exceeded the amount of esters more than two-fold, whereas in the KT line, the content of esters was higher by 20% than that of glycosides. Nevertheless, in both the control ATMA and KT root lines, sitosterol was the main sterol occurring in both ester and glycoside forms, and again stigmasterol was the second most abundant among the esters, and the less abundant among the glycosides (approx. equal with isofucosterol).

In contrast to the KT line roots, in which the total content of steroids decreased after elicitation, in the MeJA-treated ATMA roots, the total content of steroids markedly increased (by 38%). Meanwhile, the same phenomenon concerning the change in the ratio of sitosterol to stigmasterol was noticed, i.e., the decrease in the content of sitosterol parallel to a very sharp increase, almost 15-fold, in stigmasterol content. As a consequence, the ratio of sitosterol to stigmasterol changed from 28:1 to 1.8:1; however, the content of sitosterol still remained higher than that of stigmasterol. The content of other sterols, i.e., campesterol, cholesterol and isofucosterol, also increased, although less dramatically than the stigmasterol content, i.e., 1.8-fold, 2.7-fold and 3.9-fold, respectively. The content of steroid ketones increased almost two-fold, without any significant change of the proportions among individual compounds.

After elicitation with MeJA, the content of sterols occurring in both ester and glycoside forms increased (esters 2.2-fold and glycosides by 28%). However, the individual compounds displayed various patterns of change, e.g., among esters, the amounts of campesterol and sitosterol increased significantly (both almost 3-fold); the content of isofucosterol and stigmasterol increased less markedly, i.e., 2.5-fold and 1.7-fold, whereas the content of cholesterol decreased 2.5-fold. Among glycosides, the most apparent modification of the quantitative profile was a seven-fold increase in the content of stigmasterol, whereas the content of campesterol and cholesterol did not change essentially, and the content of sitosterol decreased by 33%.

The content of triterpenoids in the control ATMA root line (Table 4) was lower by 30% than in the KT root line. In both the control and elicited hairy roots, the most abundant compound was α-amyrin, constituting 46% of the total triterpenoid fraction, followed by the methyl ester of ursolic acid. As in the KT line, ursane-type triterpenoids were more abundant than the oleanane-type compounds, and free acids were the minor compounds, together constituting only 3.8% of the total triterpenoid fraction in the control roots and 6% in the elicited roots, due to the sharp (10-fold) boost of ursolic acid. Elicitation with MeJA increased the total content of triterpenoids in the ATMA root line almost six-fold, thus more significantly than in the case of the KT root line.

### 2.4. The Effect of the MeJA Treatment on Paclitaxel Content in KT and ATMA Hairy Roots

As it was reported previously [13], the two lines of *T.* × *media* hairy roots differed significantly in their capacity for paclitaxel production. The results of the present study (Table 5) confirmed the earlier report.

In the KT root line, paclitaxel was present in small amounts also in non-elicited samples; however, after elicitation with MeJA, its content increased almost six-fold. In contrast, the control ATMA roots did not contain paclitaxel at all, whereas after the MeJA treatment, the biosynthesis of this compound was intensively induced, and finally its content exceeded more than eight-fold the level detected in the elicited KT root line. It can be concluded that the elicitation step was essential for inducing taxane biosynthesis in the genetically modified hairy root line harboring the *TXS* transgene. Moreover, once induced, the paclitaxel production in the ATMA line was very efficient and significantly higher that of the KT line.

### 2.5. The Effect of the MeJA Treatment on Total Phenolic Content in KT and ATMA Hairy Roots

The changes in the total phenolic content analyzed in methanol extracts from the two lines of *T.* × *media* hairy roots after the treatment with MeJA are presented in Figure 6.

The content of phenolic compounds was slightly (by 10%) higher in the KT root line than the ATMA root line. After elicitation, it increased in both lines by approx. 30%; however, due to the considerable scattering of results, the differences were not statistically significant.

## 3. Discussion

The stimulation of the synthesis of specialized metabolites in plant in vitro cultures, which might be achieved by elicitation, often results in the allocation of metabolic resources and the redirection of the carbon flux in the metabolic pathway net, usually at the expense of the general metabolism. One of the known examples of this “growth-defense trade-off” is the competition between sterols, considered as general metabolites, and specialized triterpenoids, which share the same biosynthetic pathway upon the common precursor squalene [4]. The sterol–triterpenoid competition was investigated on numerous plant in vitro cultures producing either triterpenoids in a free form, or their glycoside derivatives, saponins [4,7,22]. The *Taxus* spp. cultures, producing the diterpene alkaloid paclitaxel, can constitute another valuable model for metabolic studies on the relations among various classes of terpenoids. Since paclitaxel and squalene-derived products theoretically share the isoprenoid pathway upon the level of sesquiterpenoids (i.e., the farnesyl diphosphate), some effects of their competition can be expected [14,15,16,17,18,23].

Although the trials of paclitaxel content enhancement involved the application of inhibitors of sterol biosynthesis [15,16,17,18], the composition of steroids in *Taxus* cultures have rarely been the main subject of investigation. Such analysis has been conducted for *Taxus chinensis* var. *mairei* suspension cultures, revealing the presence of sitosterol, campesterol and stigmasterol as the main constituents, accompanied by minor compounds as isofucosterol, clerosterol, Δ^7^-avenasterol (i.e., 24Z-ethylidene- cholest-7-en-3β-ol), ^Δ5,24(25)^-stigmastadienol, and sterol precursor, cycloartenol [19]. Thus, by comparison with the results obtained in the present study, it can be concluded that some features of the steroid composition seem to be common for *Taxus* spp. cultures, i.e., the predominance of sitosterol, campesterol and stigmasterol, and the presence of other 24-ethylsterols, such as isofucosterol. The differences among *Taxus* species can involve the occurrence of Δ^7^-steroids (not detected in *T.* × *media* in the present study); in turn, the presence or absence of biosynthetic precursors, such as cycloartenol, can be related to the type of the culture, and its developmental stage.

The two genetically different lines of *T.* × *media* hairy roots applied in the present study differed in the total content of steroids and triterpenoids (in the ATMA root line, their amounts were lower by 43% and 30%, respectively), but not in the composition of these compounds. The metabolic response to elicitation with MeJA was also different: in the KT root line, the content of steroids in free form decreased by 18%, whereas it increased by 38% in the ATMA line. However, there were several important features appearing after the MeJA treatment that were common for the two hairy root lines, i.e.,

(i)The characteristic changes in the ratio of sitosterol to stigmasterol content, caused by the decrease in the amount of sitosterol parallel to the very sharp boost in stigmasterol level;(ii)The increase in the glycoside forms of sterols, 2.5-fold in the KT line and 28% in the ATMA line (although the changes in the content of esters were diverse);(iii)The increase in the triterpenoid content (two-fold in the KT line and six-fold in the ATMA line);(iv)The slight increase (by approx. 30%) in the total phenolic content.

The increased content of stigmasterol accompanied by the decrease in sitosterol content is one of the most characteristic effects of elicitation with MeJA demonstrated in the present study. Stigmasterol is formed from sitosterol by the sterol C-22 desaturase and contains an additional double bond in the side chain at position C-22. A higher level of stigmasterol within plant plasma membrane has been correlated with altered membrane order, fluidity and permeability as well as the activation of plasma membrane H^+^-ATPase, a pump creating a pH and electrochemical gradient essential for maintaining ion homeostasis. The changes in membrane properties can further influence the activity of membrane-bound enzymes [24,25,26]. Fluctuations in the ratio of sitosterol to stigmasterol have been observed in plant responses to certain environmental stress factors, e.g., dehydration or salt stress, suggesting that the conversion of sitosterol to stigmasterol may influence plant adaptation to environmental stimuli [26]. In plant in vitro cultures, the alterations in the ratio of sitosterol to stigmasterol content were observed, for example, in *Calendula officinalis* hairy roots elicited by JA and heavy metal ions [7,22]. In a *T. chinensis* (Chinese yew) var. *mairei* suspension culture exposed to hydrodynamic mechanical (shear) stress, the enhancement of plasma membrane permeability was observed, accompanied by increased stigmasterol levels at the expense of sitosterol and campesterol content [19]. Nevertheless, the changes in sitosterol-to-stigmasterol ratio cannot be treated as an universal phenomenon occurring in plant responses to stress factors or elicitation, because various plant species greatly differ in sterol composition (e.g., species containing mainly Δ^7^-steroids). Additionally, other mechanisms, as modifications of the ratio of sterol conjugated forms, can also influence membrane properties; moreover, some plants may have distinct adaptations to stress factors, not depending directly on the modifications of sterol content in membranes [26].

Sterol esters are usually the prevailing sterol form among conjugated sterols; surprisingly, in the ATMA root line, they were the less abundant (by comparison to glycosides). However, their level significantly increased after elicitation, whereas it decreased in the KT root line. In turn, the content of sterol glycosides increased in both the KT and ATMA lines after MeJA elicitation. The changes in the ratio of conjugated to free forms of sterols is often considered a major key to modulate the membrane biophysical properties, as it can be observed, for example, in response to some environmental cues as dehydration or cold stress [24,26]. However, it is still difficult to unambiguously correlate the changes in the level of sterol forms to mechanisms of stress response, because such changes can appear as a secondary effect of the induced modifications of enzyme activity (e.g., sterol acyltransferases or glycosyltransferases) [26].

Triterpenoids are minor terpenoid compounds in conifers [27], so it is not surprising that only small amounts of the most common representatives were found in the analyzed *Taxus* hairy roots. Nevertheless, in both the KT and ATMA lines, triterpenoid content increased after elicitation, particularly high in the ATMA root line.

The results obtained in the present study confirm numerous observations that the distinct lines of plant in vitro cultures of one species, which are genetically modified or differing only in an origin of explants used for the initialization of in vitro culture, can significantly differ in the composition of general and specialized metabolites as well as in the metabolic response to elicitation [4]. For example, the two *C. officinalis* hairy root lines, differing in tissue origin (one line was derived from the cotyledon, and the other from the hypocotyl explants), displayed different sensibility and mechanisms of response to various abiotic and biotic elicitors [7,22]. Likewise, various metabolic responses to elicitation and opposed influence on steroid and triterpenoid pathways were demonstrated in the transgenic lines of the hairy roots of *Panax ginseng*, *Bupleurum falcatum* or *Centella asiatica* [4,28,29,30,31].

The elicitation with MeJA usually leads to the stimulation of many metabolic pathways and, as a consequence of modified regulatory interplay and a potential competition for biosynthetic resources, it results in deep alterations in the composition and content of specific classes of compounds. The profiling of transcripts after MeJA elicitation of in vitro cultures of various *Taxus* species demonstrated the upregulation not only of genes involved in taxane biosynthesis, but also in the transport and degradation of these compounds, as well as diverse genes engaged in cell cycle, meiosis, DNA replication, phytochrome signal transduction, general MeJA signaling network, the biosynthesis of plant hormones and some specialized metabolites, such as phenylpropanoids [9,13,32,33].

The main effects of elicitation with MeJA observed in the present study are summarized in Figure 7.

In the MeJA-elicited KT root line, the enhancement of terpenoid volatiles and retinol amounts was observed, combined with the decrease in sterol content, which suggest the stimulation of terpenoid pathways leading to mono-, sesqui- and diterpenoids (and further to tetraterpenoids, e.g., carotenoids). On the level of squalene cyclization, the typical symptoms of a competition between sterols and triterpenoids were observed, although the stimulation of the triterpenoid content was rather slight (only two-fold). In contrast, in the MeJA-elicited ATMA root line, the enhancement of mono- and sesquiterpenoids was not detected. The presence of increased levels of retinol and retinol acetate might suggest the stimulation of diterpenoid and tetraterpenoid biosynthesis. The competition at the level of squalene cyclization did not appear, resulting in the simultaneous enhancement of sterols and triterpenoids.

The enhancement of the mono- and sesquiterpenoid pathways in a KT line that is not genetically modified can suggest that the production of volatile compounds of these two classes, which are typical for conifer species, might be a part of the normal physiological response to MeJA elicitation in *Taxus* plants. These compounds are often considered as defense metabolites, acting in environmental signaling, but also exerting antiherbivore and antimicrobial activity. In contrast, this phenomenon did not occur in the ATMA line, pointing to the substantial change in terpenoid pathway imposed by the presence of the taxadiene synthase (*T**XS)* transgene. It seems that this type of genetic modification allows to omit the stimulation of early branches in terpenoid pathways, and instead, to induce further biosynthetic steps, leading to di- and triterpenoids.

However, it seems unlikely that, even regarding the demonstrated enhancement of sterol and triterpenoid content in the elicited ATMA root line, the biosynthetic pathway leading to squalene in *Taxus* in vitro cultures is a main competitor that limits paclitaxel production. Sterols and free triterpenoids are located mainly in membranes; therefore, even the targeted trials of their enhanced production in plant in vitro cultures appear problematic [4,24,34]. Therefore, the strategy of the inhibition of steroid and triterpenoid biosynthesis does not seem to be essential in the biotechnological production of paclitaxel derived from *Taxus* in vitro cultures. This conclusion is additionally supported by the present results concerning the ATMA root line, in which the observed increase in the sterol content was parallel to the very intensive enhancement of paclitaxel biosynthesis. In turn, the MeJA-stimulated biosynthesis of phenolic compounds demonstrated in the present study was relatively low (approx. by 30% in both lines). Thus, the stimulation of the phenylpropanoid pathway seems to be much weaker in *Taxus* cultures than in other plant hairy root cultures, e.g., *Lactuca indica* [35], *Ajuga bracteosa* [36], or *Salvia miltiorrhiza* [37], showing the total phenolic content to be more than two-fold higher after MeJA elicitation in comparison to the control. Therefore, the strategy of precursor feeding with external phenylalanine, applied in some studies on paclitaxel production improvement [13,21], might be reasonable and potentially more advantageous. However, the strategy of the inhibition of steroid and triterpenoid biosynthesis might be of greater importance in microbial cultures metabolically and genetically engineered for paclitaxel production, as it was demonstrated in some reported studies on various fungi strains [12,38,39].

## 4. Materials and Methods

### 4.1. Plant Material

Experiments were performed on two lines of *T.* × *media* hairy roots, KT and ATMA. The KT line was obtained by the transformation of 8-week-old seedlings with the *Agrobacterium tumefaciens* strain LBA 9402 [20], whereas the ATMA line was developed as a result of transformation of 10-year-old *T.* × *media* plantlets cultivated in vitro with the *A. tumefaciens* C58C1 strain [21], which carries the taxadiene synthase (*T**XS)* transgene from *T. baccata* (GenBank accession: AY424738). The hairy roots were maintained in 250 mL Erlenmeyer flasks containing 35 mL of a hormone-free liquid DCR medium modified by an increased concentration of MgSO_4_ (400 mg/L) (DCR-M) [40], at 23 ± 1 °C in the dark, on the INFORS AG TR 250 shaker (Bottmingen, Switzerland) operating at 105 rpm, and routinely sub-cultured every 4 weeks.

### 4.2. Elicitation of Hairy Root Cultures

Both lines of hairy roots were treated with methyl jasmonate (MeJA, Sigma-Aldrich, Steinheim, Germany) at a final concentration of 100 μM, added to the media under sterile conditions in a laminar cabinet. In all experiments (i.e., control and elicitation), inoculum in a form of 0.5 ± 0.05 g of 28-day-old roots was transferred to the fresh DCR-M medium. Afterwards, the elicited samples were supplemented with MeJA on the 28th day of culture. The samples were then harvested on the 14th day of culture. The biomass of hairy roots were then separated from the medium, gently pressed on filter paper, weighed to obtain fresh weight (FW) and lyophilized (lyophilizer Christ ALPHA1-4 LSC, Osterode am Harz, Germany) to obtain the dry weight (DW).

### 4.3. Extraction of Hairy Roots

The lyophilized hairy roots (sample masses ranging from 0.151 to 0.335 mg) were powdered in a laboratory mortar, poured into glass bottles (200 mL volume) and extracted first with 3 subsequent portions of diethyl ether (dried material-to-solvent ratio of 1:200, g:mL) and then with 3 portions of methanol (the same root powder-to-solvent ratio as for diethyl ether extraction, i.e., 1:200) with the use of an ultrasonic water bath Emmi-40HC (EMAG Technologies, Mörfelden-Walldorf, Germany) at a frequency of 40 kHz, power of 250 W, at the initial temperature of 20 °C, and the final temperature maintained below 38 °C. In each extraction cycle, the extraction lasted 20 min, and afterwards, the extracts were filtered through Whatman No. 1 filter paper; the solid residues were put back in the bottles and subjected to subsequent extraction. The obtained extracts (separately. the combined three diethyl ether extracts and the combined three methanol extracts) were evaporated to dryness under reduced pressure on a rotary evaporator Laborota 4000 (Heidolph Instruments, Schwabach, Germany). The general scheme of the experimental procedure is presented in Appendix A. The diethyl ether extract masses of the initial 250 mg sample of the hairy roots obtained in the subsequent steps of UAE (ultrasound-assisted extraction) were: 7.6 ± 0.1, 3.5 ± 0.08 and 1.8 ± 0.06 mg for the KT root line (12.9 mg in total), and 7.4 ± 0.1, 3.3 ± 0.08 and 1.6 ± 0.05 mg for the ATMA root line (12.3 mg in total). The methanol extract masses were 63.2 ± 0.94, 9.8 ± 0.11 and 4.3 ± 0.08 mg for the KT root line (77.3 mg in total), and 62.4 ± 0.96, 9.5 ± 0.1 and 4.1 ± 0.06 mg for the ATMA line (76.0 mg in total). The masses of extracts obtained with diethyl ether in the Soxhlet apparatus produced 12.6 ± 0.12 mg and 12.15 ± 0.11 mg for the KT and ATMA root lines, respectively, and the masses of methanol extracts were 75.8 ± 1.2 mg and 73.7 ± 1.0 mg. Thus, the extraction yield calculated according to extract mass was comparable for the two methods of extraction.

To calculate the total recovery and to compare the yield of UAE with extraction in Soxhlet apparatus, additional experiments on the samples of the control KT line hairy roots were made, by adding the known quantity (1 mg) of 5β-cholestan-3α-ol (Sigma-Aldrich, Steinheim, Germany) to 250 mg samples of roots prior to their extraction either by UAE (as described above) or using a Soxhlet apparatus for 8 h with diethyl ether and then 8 h with methanol. The obtained extracts were fractionated and analyzed as all the others, according to subsequent steps of the experimental procedure.

### 4.4. Fractionation of Diethyl Ether Extracts

Diethyl ether extracts were fractionated by adsorption preparative TLC on 20 cm × 20 cm glass plates, manually coated with silica gel 60H (Merck, Darmstadt, Germany). The solvent system chloroform: methanol 97:3 (*v*/*v*) was used to develop the plates. Four fractions were obtained: (i) esters (Rf, retention factor, of 0.9–1); (ii) free steroids and neutral triterpenoids (Rf of 0.3–0.9); (iii) free triterpenoid acids (Rf 0.2–0.3); and (iv) glycosides (Rf of 0–0.2). Fraction (ii) containing free steroids and neutral triterpenoids (alcohols) was directly analyzed using GC–MS, gas chromatography-mass spectrometer (Agilent Technologies 7890A); the triterpenoid acid fraction (iii) was methylated with diazomethane prior to GC-MS analysis; the ester fraction (i) was subjected to alkaline hydrolysis; and the glycoside fraction (iv) to acidic hydrolysis.

### 4.5. Alkaline Hydrolysis

The ester fractions were subjected to alkaline hydrolysis with 10% NaOH in 80% methanol at 80 °C for 3 h. Subsequently, 5 volumes of water were added to each hydrolysate, the pH was neutralized with 5% acetic acid, and the obtained mixtures were extracted with diethyl ether (3 × 20 mL) in a separation funnel. The extracts were evaporated and fractionated by preparative TLC (as described in Section 4.4) to obtain the fraction of sterols released from their ester forms.

### 4.6. Acidic Hydrolysis

The glycoside fractions obtained from the diethyl ether extracts and parts (halves) of methanol extracts were hydrolyzed by 11% HCl in 80% methanol during 2 h on a heating mantle under reflux [41]. Subsequently, the hydrolysates were diluted with distilled water, methanol was evaporated in a rotary evaporator, and the obtained aqueous remnants were extracted 3 times with 40 mL portions of diethyl ether in a separation funnel. The obtained extracts were washed with distilled water 3 times and evaporated to dryness, then fractionated by TLC (Section 4.4) to obtain the fraction of sterols released from their glycoside forms.

### 4.7. Identification and Quantification of Steroids by GC-MS

The qualitative and quantitative analyses of steroid and triterpenoid fractions were made with the use of an Agilent Technologies 7890A gas chromatograph (GC–MS) (Perlan Technologies, Warszawa, Poland) equipped with a 5975C mass selective detector, on a 30 m × 0.25 mm i.d. (internal diameter) column, 0.25-μm, HP-5MS UI (Agilent Technologies, Santa Clara, CA, USA). Samples dissolved in diethyl ether:methanol (5:1, *v*/*v*) were applied using 1:3 split injection. Helium was used as the carrier gas at a flow rate of 1 mL/min. The separation was made with the following temperature program: an initial temperature of 160 °C held for 2 min, then increased to 280 °C at 5 °C/min, and a final temperature of 280 °C held for a further 44 min. The other employed parameters were as follows: inlet and FID (flame ionization detector) temperature of 290 °C; MS transfer line temperature of 275 °C; quadrupole temperature of 150 °C; ion source temperature of 230 °C; EI of 70 eV; *m*/*z* range of 33–500; FID gas (H2) flow of 30 mL/min (hydrogen generator HydroGen PH300, Peak Scientific, Inchinnan, UK); and airflow of 400 mL/min. Individual compounds were identified by comparing their mass spectra with library data from Wiley 9th ED. and NIST 2008 Lib. SW Version 2010, or previously reported data, and by comparison of their retention times and corresponding mass spectra with those of authentic standards, where available. The quantitation of steroids was performed using an external standard method based on calibration curves determined for an authentic standards of sitosterol (Sigma-Aldrich, Steinheim, Germany), α-amyrin and ursolic acid methyl ester (Roth, Karlsruhe, Germany).

The total recovery of a known quantity of 5β-cholestan-3α-ol (Sigma-Aldrich, Steinheim, Germany), not present in the analyzed samples, added as an internal standard to KT root samples (Figure 2, KT control, Section 4.3) prior to their extraction, was calculated as 96% for UAE and 94% for extraction in Soxhlet apparatus. Thus, the total recovery was comparable to the total recovery calculated for triterpenoid faradiol (96%), estimated previously for the extraction in a Soxhlet apparatus during 8 h [42].

### 4.8. Determination of Paclitaxel Content

The samples of the lyophilized roots (100 mg) were powdered and extracted with 1 mL of methanol using an ultrasonic bath (Sonorex Bandelin, Berlin, Germany) during 15 min at a frequency of 35 kHz, power of 250 W, and the temperature maintained below 40 °C. Afterwards, the samples were shaken overnight in the dark. Next, after centrifugation for 15 min at 15,500 rpm and at 5 °C, the supernatants were collected. The residues were reextracted with 1 mL portions of methanol and ultrasonicated for 15 min. The supernatants were combined and evaporated to dryness. The dry residue was reconstituted in 1 mL of methanol, and 0.5 mL of this mixture was brought to 5 mL of water and loaded on the Solid Phase Extraction (SPE) cartridge (Supelco, Bellfonte, PA, USA). Extracts were cleaned by SPE on Supelclean LC-18 tubes (Supelco Sigma-Aldrich, St. Louis, MO, USA) by washing with water, 20% methanol, 50% methanol and finally 100% methanol, according to the method described earlier [20]. The fractions eluted with 100% methanol were collected, evaporated to dryness, re-dissolved in methanol and then underwent High Performance Liquid Chromatography with Diode Array Detection (HPLC–DAD) analysis using the DIONEX (Sunnyvale, CA, USA) system with a UVD 340S diode array detector and an automated sample injector (ASI-100) employing a gradient of 0.05 M ammonium acetate/acetonitrile (7:3 *v*/*v* and 1:9 *v*/*v*) with a flow rate of 0.8 mL/min, as described previously [13]. The peak of paclitaxel was assigned by spiking the samples with the standard and comparing the retention times and UV spectra. The paclitaxel was identified and quantified at 227 nm. The standard compound was produced by CHROMADEX Longmonton, CO, USA) and purchased from LGC Standards (Teddington, UK). All chemicals for HPLC analysis were purchased from Sigma-Aldrich (Poznań, Poland).

### 4.9. Total Phenolic Content

The TPC in the obtained methanol extracts was determined spectrophotometrically using Folin–Ciocalteu’s method. Half of the obtained extracts were air-dried and dissolved in 5 mL of methanol. For the reaction, 3 μL samples were mixed with deionized water (247 μL) and Folin–Ciocalteu reagent (250 μL). After 3 min, 500 μL of 25% *w*/*v* anhydrous sodium carbonate was added to each sample and mixed thoroughly. The mixture was kept in the dark for 1 h at room temperature. The absorbance was measured at 750 nm (UV 2401PC Spectrophotometer, Shimadzu, Kyoto, Japan). All results are expressed as the gallic acid equivalent (mg/g D.W.) calculated from interpolating the respective absorbance in the calibration curve.

### 4.10. Statistical Analysis of Data

All experiments were performed in triplicate. Data are presented as the means ± standard error of three independent samples. The data were subjected to one-way analysis of variance (ANOVA), and the differences between means were evaluated using Duncan’s multiple-range test. Statistical significance was considered to be obtained at *p* < 0.05.

## 5. Conclusions

The present study is the first report on the modifications of the terpenoid biosynthetic pathway in *Taxus* hairy root cultures triggered by MeJA, particularly concerning steroids and triterpenoids. The obtained results reveal that two genetically different hairy root lines of *T.* × *media* did not vary in the composition of these compounds, but in their quantitative profile and in the changes of their content induced by elicitation with MeJA. The main demonstrated difference between the two lines was the opposite influence on sterol metabolism, i.e., the decrease in the content of steroids in a free form by 18% in the KT root line versus their increase by 38% in the ATMA line. Moreover, the elicitation with MeJA seemed to stimulate the terpenoid pathways leading to mono-, sesqui- and diterpenoids in the KT line, whereas to di- and triterpenoids in the ATMA line. However, there were also several important metabolic features common for the two hairy root lines, including the characteristic changes in the ratio of sitosterol to stigmasterol content, caused by the decrease in the amount of sitosterol parallel to the very sharp boost in stigmasterol level, the increase in the amount of glycoside forms of sterols, as well as in triterpenoid and total phenolic content. The obtained results contribute to a better understanding of the relations between general and specialized metabolism in elicited plant in vitro cultures, and might be useful in designing the strategies of their metabolic engineering.

## Figures and Tables

**Figure 1 plants-11-01120-f001:**
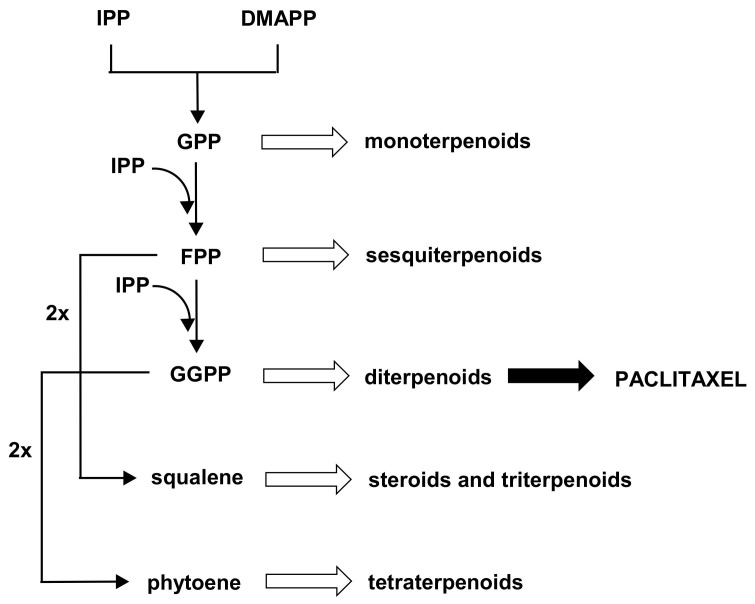
The scheme of the general pathway of terpenoid biosynthesis, regardless of the MVA (mevalonate) or MEP (methylerythritol phosphate) origin of precursors. IPP, isopentenyl diphosphate; DMAPP, dimethylallyl diphosphate; GPP, geranyl diphosphate; FPP, farnesyl diphosphate; GGPP, geranylgeranyl diphosphate.

**Figure 2 plants-11-01120-f002:**
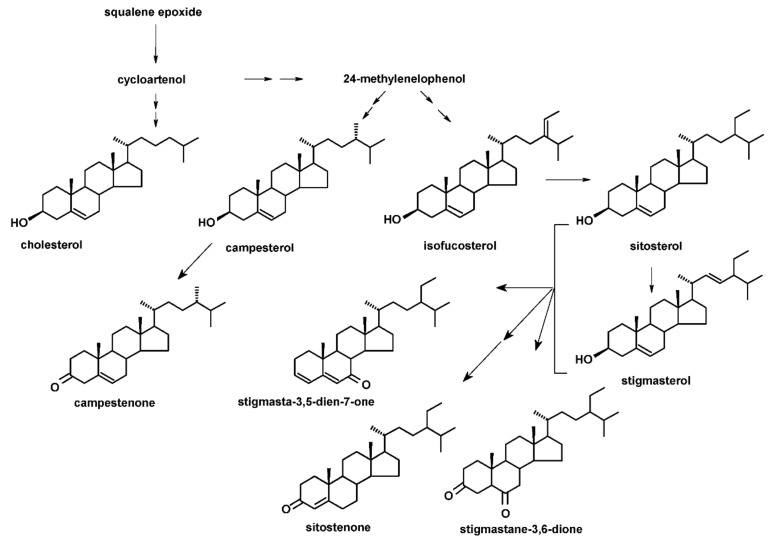
The structures of the steroids identified in *T.* × *media* hairy roots organized in simplified biosynthetic pathways (precursors and important branching points are incorporated in the scheme only as names). Three steroid ketones (stigmasta-3,5-dien-7-one, sitostenone and stigmastane-3,6-dione) can be derived from both sitosterol and stigmasterol.

**Figure 3 plants-11-01120-f003:**
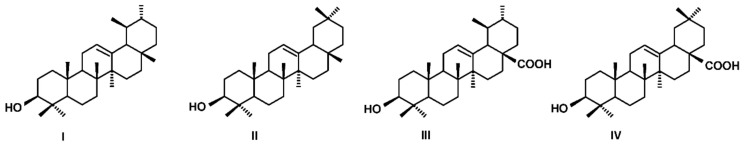
The structures of the triterpenoids identified in *T.* × *media* hairy roots. I, α-amyrin; II, β-amyrin; III, ursolic acid; IV, oleanolic acid.

**Figure 4 plants-11-01120-f004:**
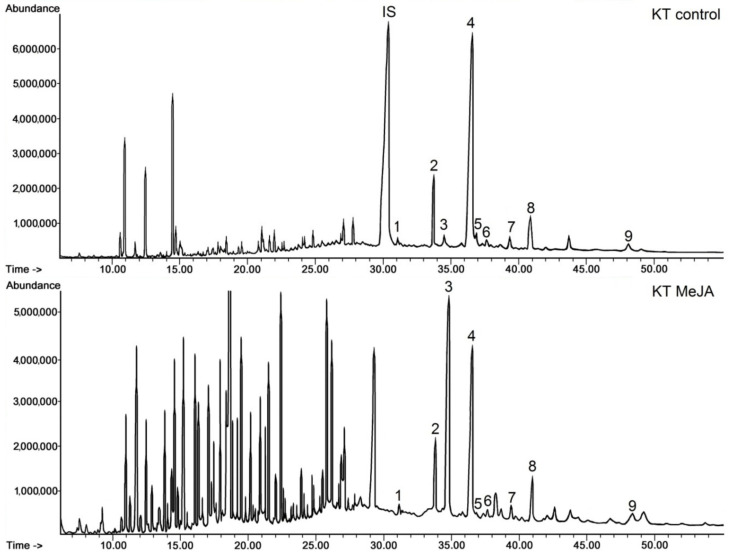
Representative chromatograms of the fractions containing free steroids from diethyl ether extracts obtained from control and elicited KT line hairy roots. 1, cholesterol; 2, campesterol; 3, stigmasterol; 4, sitosterol; 5, isofucosterol; 6, campestanol; 7, tremulone; 8, sitostenone; 9, stigmastanedione; IS, internal standard (5β-cholestan-3α-ol).

**Figure 5 plants-11-01120-f005:**
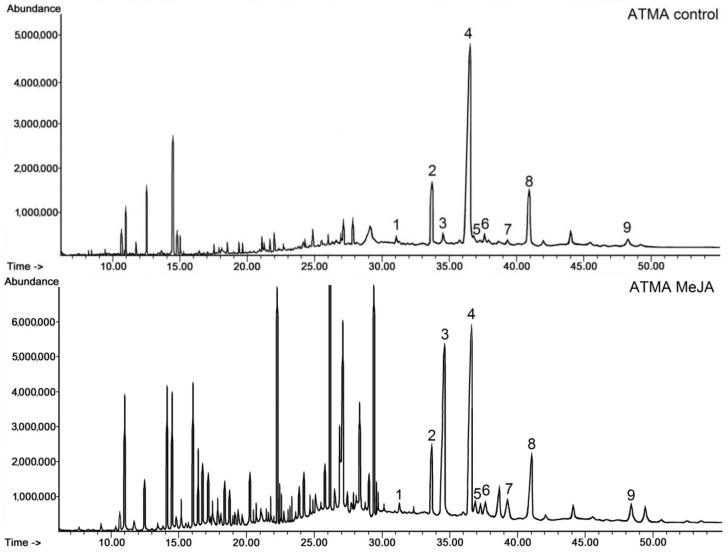
Chromatograms of the fractions containing free steroids from diethyl ether extracts obtained from the control and elicited ATMA line hairy roots. 1, cholesterol; 2, campesterol; 3, stigmasterol; 4, sitosterol; 5, isofucosterol; 6, campestanol; 7, tremulone; 8, sitostenone; 9, stigmastanedione.

**Figure 6 plants-11-01120-f006:**
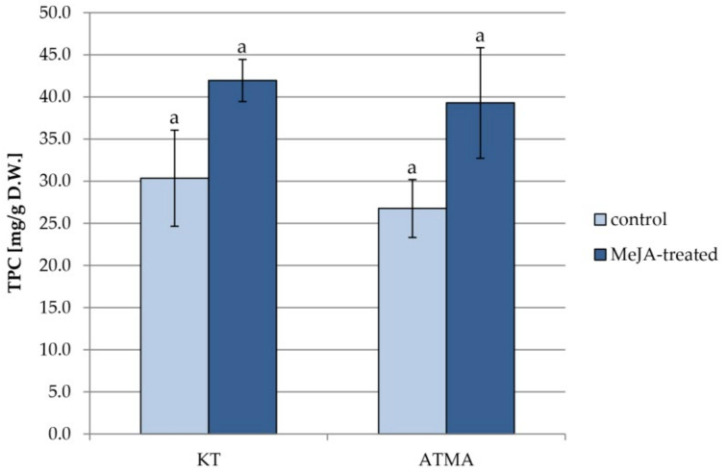
The total phenolic content (TPC) in methanol extracts obtained from control and elicited hairy roots of KT and ATMA lines. Results not sharing a common letter are significantly different (*p* < 0.05).

**Figure 7 plants-11-01120-f007:**
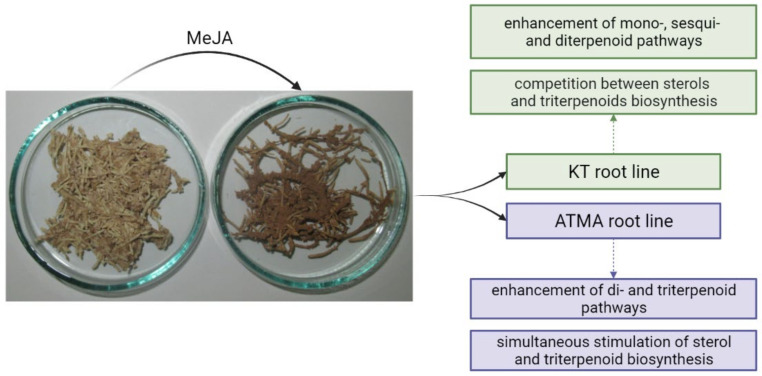
The main metabolic changes occurring in the terpenoid pathway in the KT and ATMA lines of *T.* × *media* hairy roots after elicitation with MeJA. The included picture (representative samples of the control and MeJA-treated ATMA root line) shows the visible difference between the control and elicited hairy roots.

**Table 1 plants-11-01120-t001:** The content of steroids in the *T.* × *media* KT hairy root line after elicitation with 100 µM MeJA.

Compound	Content (µg/g D.W. ± SE)					
	Free Forms		Esters		Glycosides	
	Control	MeJA-Treated	Control	MeJA-Treated	Control	MeJA-Treated
campesterol	185.73 ± 5.77 a	152.39 ± 9.32 b	1.74 ± 0.08 a	0.73 ± 0.06 b	2.86 ± 0.11 a	4.57 ± 0.05 b
cholesterol	33.02 ± 4.12 a	13.10 ± 1.92 b	0.64 ± 0.05 a	0.44 ± 0.03 b	2.74 ± 0.10 a	5.16 ± 0.05 b
isofucosterol	47.52 ± 4.18 a	9.14 ± 1.58 b	2.07 ± 0.09 a	1.12 ± 0.08 b	2.01 ± 0.07 a	3.18 ± 0.11 b
sitosterol	1319.23 ± 61.29 a	546.00 ± 18.20 b	9.94 ± 0.72 a	5.16 ± 0.19 b	4.27 ± 0.12 a	8.33 ± 0.36 b
stigmasterol	61.50 ± 5.08 a	627.85 ± 40.45 b	2.35 ± 0.05 a	2.30 ± 0.08 a	1.55 ± 0.04 a	12.34 ± 0.46 b
campestenone	31.15 ± 3.88 a	22.57 ± 0.42 b				
sitostenone	120.01 ± 10.21 a	132.45 ± 2.10 a				
stigmastanedione	67.58 ± 5.96 a	34.49 ± 1.19 b				
tremulone	61.71 ± 4.66 a	39.94 ± 6.25 b				
*Sum of sterols*	*1646.99*	*1348.47*				
*Sum of ketones*	*280.45*	*229.46*				
Total	1927.44	1577.93	16.74	9.75	13.43	33.58

Results refer to hairy root dry weight and expressed as the mean ± SE of three independent samples. Results are compared statistically within categories (free forms, esters and glycosides); results not sharing a common letter are significantly different (*p* < 0.05).

**Table 2 plants-11-01120-t002:** The content of triterpenoids in the *T.* × *media* KT hairy root line after elicitation with 100 µM MeJA.

Compound	Content (µg/g D.W. ± SE)	
	Control	MeJA-Treated
α-amyrin	17.21 ± 0.57 a	31.82 ± 2.86 b
β-amyrin	4.51 ± 0.18 a	9.30 ± 0.58 b
oleanolic acid	1.00 ± 0.03 a	3.14 ± 0.09 b
ursolic acid	3.29 ± 0.08 a	8.02 ± 0.08 b
ursolic acid methyl ester	27.51 ± 3.68 a	48.93 ± 2.49 b
*sum of amyrins*	*21.72*	*41.12*
*sum of acids*	*31.80*	*60.09*
Total	53.52	101.21

Results refer to hairy root dry weight and expressed as the mean ± SE of three independent samples. Results not sharing a common letter are significantly different (*p* < 0.05).

**Table 3 plants-11-01120-t003:** The content of steroids in the *T.* × *media* ATMA (harboring the *TXS* transgene) hairy root line after elicitation with 100 µM MeJA.

Compound	Content (µg/g D.W. ± SE)					
	Free Forms		Esters		Glycosides	
	Control	MeJA-Treated	Control	MeJA-Treated	Control	MeJA-Treated
campesterol	108.42 ± 5.51 a	194.34 ± 5.63 b	0.95 ± 0.05 a	2.78 ± 0.07 b	3.56 ± 0.08 a	3.62 ± 0.03 a
cholesterol	10.12 ± 0.54 a	27.02 ± 1.65 b	0.99 ± 0.04 a	0.47 ± 0.07 b	4.60 ± 0.10 a	4.67 ± 0.05 a
isofucosterol	12.19 ± 0.69 a	47.02 ± 3.78 b	0.75 ± 0.05 a	1.84 ± 0.05 b	1.63 ± 0.07 a	1.41 ± 0.04 b
sitosterol	695.44 ± 36.87 a	650.58 ± 18.21 a	4.20 ± 0.13 a	11.58 ± 0.48 b	10.35 ± 0.72 a	7.98 ± 0.15 b
stigmasterol	24.55 ± 2.73 a	364.26 ± 26.39 b	2.43 ± 0.08 a	4.05 ± 0.13 b	1.64 ± 0.06 a	11.71 ± 0.19 b
campestenone	23.08 ± 2.05 a	50.09 ± 6.58 b				
sitostenone	140.42 ± 11.08 a	268.06 ± 10.18 b				
stigmastanedione	37.86 ± 4.66 a	71.12 ± 6.19 b				
tremulone	34.79 ± 4.48 a	67.05 ± 5.22 b				
*Sum of sterols*	850.72	1283.22				
*Sum of ketones*	236.16	456.33				
Total	1086.88	1739.55	9.32	20.72	21.78	29.39

Results refer to hairy root dry weight and expressed as the mean ± SE of three independent samples. Results are compared statistically within categories (free forms, esters and glycosides); results not sharing a common letter are significantly different (*p* < 0.05).

**Table 4 plants-11-01120-t004:** The content of triterpenoids in the *T.* × *media* ATMA hairy root line after elicitation with 100 µM MeJA.

Compound	Content (µg/g D.W. ± SE)	
	Control	MeJA-Treated
α-amyrin	17.48 ± 1.46 a	102.88 ± 5.49 b
β-amyrin	8.78 ± 0.75 a	42.83 ± 2.31 b
oleanolic acid	0.32 ± 0.02 a	1.45 ± 0.05 b
ursolic acid	1.13 ± 0.04 a	11.35 ± 0.22 b
ursolic acid methyl ester	10.04 ± 0.45 a	62.54 ± 3.58 b
*sum of amyrins*	*26.26*	*145.71*
*sum of acids*	*11.49*	*75.35*
Total	37.75	221.05

Results refer to hairy root dry weight and expressed as the mean ± SE of three independent samples. Results not sharing a common letter are significantly different (*p* < 0.05).

**Table 5 plants-11-01120-t005:** The content of paclitaxel in the *T.* × *media* KT and ATMA hairy root lines after elicitation with 100 µM MeJA.

Compound	Content (µg/g D.W. ± SE)			
	KT Line		ATMA Line	
	Control	MeJA-Treated	Control	MeJA-Treated
paclitaxel	31.7 ± 6.3 a	187.8 ± 11.2 b	0	1532.8 ± 28.9 a

Results refer to hairy root dry weight and expressed as the mean ± SE of three independent samples. Results not sharing a common letter are significantly different (*p* < 0.05).

## Data Availability

Not applicable.

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
