# Peer review of "Metabolic Modifications in Terpenoid and Steroid Pathways Triggered by Methyl Jasmonate in Taxus × media Hairy Roots"

_plants, 2022, doi:10.3390/plants11091120_

Round 1

Reviewer 1 Report

The present study evaluates the impact of MJ treatment on the steroid and triterpenoid content in hairy root cultures of Taxus x media. The formal side of the manuscript corresponds to the instructions of the journal. Regarding the content, I recommend minor modifications in the Discussion listed below.
Paragraph 1
Authors suppose a possible competition between biosynthetic pathways of paclitaxel and squalene-derived products. In this relation, it could be desirable to give some information about the content of paclitaxel in MJ treated hairy root cultures of Taxus x media examined in the present study. These data would complete the overall picture of metabolic changes in hairy root cultures under elicitation.

Author Response

Initially we did not intend to include the results on paclitaxel content in this manuscript, because the main aim was different, the results on steroid and triterpenoid content are new and never studied before, whereas the effect of MeJA elicitation on paclitaxel production in the studied hairy root lines have been already reported. However, we respect the Reviewer’s opinion and we agree that these data are valuable to complete the presented picture of metabolic changes. Since the paclitaxel analysis was done routinely in this experiment to confirm the metabolic stability of the root lines, we could insert the required results in the present manuscript. Finally we appreciate the Reviewer’s demand because with these results the conclusion is even better supported (we have added one sentence to the discussion).

Reviewer 2 Report

In the manuscript entitled “Metabolic Modifications in Terpenoid and Steroid Pathways Triggered by Methyl Jasmonate in Taxus x media Hairy Roots" authored by Katarzyna SykÅ‚owska-Baranek et al., 

the authors present in an eloquent and clear manner the results of the study. The manuscript does not require editing of English language and style.

The methods and the protocols are standardized and suitable for the study aim.

The importance and novelty of the research are well underlined

Still, there is a recommendation regarding the statistical analysis that should be improved  - t-Student’s test is useful for a basic evaluation, but ANOVA is recommended

Author Response

Thank you for the comments. We have change the method of the statistical analysis for ANOVA.

Reviewer 3 Report

Queries

  1. “from diethyl ether extracts” (P 4)

“in methanol extracts” (P 9)

“diethyl ether……methanol” (P 13)

What are the different?

Please provide the scheme of extraction processes and address clearly.

Please provide the moisture and weight of matrix.

Please provide the final temperature of each extraction step.

Please also provide the yields in each step.

  1. What are the different results between UAE and Soxhlet extraction in compositions? (P 13)

Comments

  1. Please be precise for the “Keywords”.

Six “Keywords” are suggested.

  1. Please revise “Taxus in vitro cultures……” to be “The in vitro cultures of Taxus……”. (Abstract)

  1. “SykÅ‚owska”, “Syklowska”, “steroid and triterpenoid”, “steroids and triterpenoids”, “Taxus x media”. “Taxus x media”. “ x media” …….

Please present the manuscript consistently.

  1. After the abbreviations are defined, please use the abbreviation throughout the manuscript.

  1. There are some typing and grammar errors. Please check carefully.

  1. Please check the reference 23

Author Response

Query 1. The scheme of the experimental procedure has been added to the Supporting Material. It clarifies the various steps of the applied method and helps to distinguish various extracts (diethyl ether and methanol) obtained in the experiment. Two-step extraction is useful to separate the free forms of steroids and triterpenoids from their conjugate forms.

The roots were lyophilized to dryness prior to their extraction. We did not measure their moisture. We are providing the weight of the consecutive extracts (resulting from each step of the UAE) and the weight of extracts obtained after extraction with the use of the Soxhlet apparatus (these data are included in the text). It is a way of comparing the yield of each step of extraction (we cannot measure the yield on each step with the use of the applied internal standard because it needs to make chromatographic fractionation before GC-MS analysis (the obtained extracts cannot be directly analyzed by GC-MS).

After each step of UAE, the remaining root mass were air-dried to obtain dry powder (we did not measure the moisture). The temperature was measured in each step of UAE, it did not exceed the 38 °C in the final step (this information was included in the text). No difference in the composition of steroids and triterpenoids, and no visible difference in other classes of the compounds appearing on the obtained chromatograms was noticed.

Query 2.

  1. The number of keywords was reduced to six.
  2. The suggested change was made both in the Abstract and Introduction.
  3. We apologize for the mess with various forms of Taxus x media, it was finally corrected either to Taxus x media when used for the first time in the Abstract and in the Introduction, afterwards the abbreviation T. x media was applied. The problem is with the titles of the chapters, which should be in italics, so we could not keep the “x” in normal writing style. In the name SykÅ‚owska there is a Polish letter “Å‚”, which was sometimes neglected in the previous publications, however, the right form is “SykÅ‚owska”. We have corrected the use of “steroids and triterpenoids”. They are used as “steroids and triterpenoids” when they are mentioned as the subject in the sentence (i.e. as the name of the type of the particular compounds), however, when they are used as adjective (e.g., steroid and triterpenoid content), they are used in singular, not plural form.
  4. The abbreviations (e.g., T. x media, MeJA, GC-MS) were used in the text after their first explanation (with the exception of T. x media which is repeated in the full form in the Abstract and Introduction).
  5. The text was rewritten and the noticed mistakes were corrected.
  6. The references were checked and corrected.

Round 2

Reviewer 3 Report

Comments

  1. There are several errors Please check carefully.
  2. “The in vitro cultures……”.
  3. Please provide the extraction parameters clearly.
  4. Please check reference 23 ……Agrobacterium rhizogenes……

Author Response

  1. We have carefully corrected the text, all new improvements are marked in green.

  1. The phrase “in vitro” is not written in italics according to the style applied in MDPI journals.

  1. We have provided all the extraction parameters like the temperature, duration, the number of cycles, the frequency and the power of UAE equipment, the yield, the total recovery, we have also added the material-to-solvent ratio.

  1. We have corrected the Latin name of Agrobacterium, we apologize for this mistake omitted in the previous revised version (when we have only checked the reference list).